# Enhanced Root-MUSIC Algorithm Based on Matrix Reconstruction for Frequency Estimation

**DOI:** 10.3390/s23041829

**Published:** 2023-02-06

**Authors:** Yingjie Zhu, Wuxiong Zhang, Huiyue Yi, Hui Xu

**Affiliations:** 1Key Laboratory of Science and Technology on Micro-System, Shanghai Institute of Microsystem and Information Technology Chinese Academy of Sciences, Shanghai 200050, China; 2University of Chinese Academy of Sciences, Beijing 100049, China

**Keywords:** root-MUSIC, frequency estimation, sparse, matrix reconstruction, FMCW radar

## Abstract

In recent years, frequency-modulated continuous wave (FMCW) radar has been widely used in automatic driving, settlement monitoring and other fields. The range accuracy is determined by the estimation of the signal beat frequency. The existing algorithms are unable to distinguish between signal components with similar frequencies. To address this problem, this study proposed an enhanced root-MUSIC algorithm based on matrix reconstruction. Firstly, based on the sparsity of a singular value vector, a convex optimization problem was formulated to identify a singular value vector. Two algorithms were proposed to solve the convex optimization problem according to whether the standard deviation of noise needed to be estimated, from which an optimized singular value vector was obtained. Then, a signal matrix was reconstructed using an optimized singular value vector, and the Hankel structure of the signal matrix was restored by utilizing the properties of the Hankel matrix. Finally, the conventional root-MUSIC algorithm was utilized to estimate the signal beat frequency. The simulation results showed that the proposed algorithm improved the frequency resolution of multi-frequency signals in a noisy environment, which is beneficial to improve the multi-target range accuracy and resolution capabilities of FMCW radar.

## 1. Introduction

Radar is an electronic device that uses electromagnetic waves to detect targets. According to the working mode of radar, radar usually has two basic types: continuous wave (CW) and pulse. Continuous wave (CW) radar is a relatively low-complexity radar system and uses only the difference between the carrier frequencies of the transmitter and receiver to estimate the velocity of the target. However, CW radar cannot measure the distance to the target [1]. Frequency-modulated continuous wave (FMCW) radar is a continuous wave radar that uses a specific signal to modulate the frequency of the transmitted signal. The signal processing of FMCW radar is performed in a low frequency band after mixing, and thus, FMCW radar systems are capable of estimating the range, Doppler, and angle of their targets. FMCW radar has the characteristics of large bandwidth, low power consumption, small size and high precision [2], and it has been widely used in military and civil applications [3,4].

In an FMCW system, the range/Doppler migration and the velocity ambiguity are two well-known problems encountered during high-speed moving target detection [5]. Therefore, the range of the targets cannot be determined by a simple beat frequency estimate. When there is no range migration, i.e., the target movement within the coherent processing interval (CPI) is less than the range resolution bin, the range of the targets are determined by a simple beat-frequency estimate.

Therefore, the estimation accuracy of the signal beat frequency determines the range accuracy of FMCW radar [6]. When there are multiple targets, the signal beat frequency is a multi-frequency signal. There are basically two ways to estimate the frequencies of a multi-frequency signal: time-domain methods and frequency-domain methods [7,8]. The frequency-domain methods are based on the fast Fourier transform (FFT) [9,10]. Due to the problems of fence effect and spectrum leakage, the frequency estimation accuracy of the FFT algorithm is low, and frequencies that are very close cannot be distinguished. The chirp-Z transform (CZT) algorithm is an improved algorithm of the FFT algorithm [11]. It can refine some frequency bands to improve the accuracy of frequency estimation. The CZT algorithm has the advantages of high frequency estimation accuracy and good anti-noise performance, but it still has the problem of fence effect, such as the FFT algorithm, and its frequency estimation accuracy is determined by multiple refinements. Time-domain methods include the linear prediction method [12], subspace-based algorithms [4,13,14,15,16], and so on. The subspace-based algorithms include the multiple signal classification (MUSIC) algorithm [13,14,15], the estimating signal parameters via rotational invariance techniques (ESPRIT) algorithm [4,16], etc., which use the relationship between the signal subspace and frequency to estimate frequency. Compared with other algorithms, the subspace-based algorithms haveigher frequency resolution ability and higher estimation accuracy for multi-frequency signals. Subspace-based algorithms also have some disadvantages, such as poor anti-noise performance and high computation complexity. In [17], the root-MUSIC algorithm was utilized to estimate the frequency of harmonic signals. Unfortunately, the frequency estimation accuracy of this method was affected by noise. In order to reduce the influence of noise on the frequency estimation accuracy, ref [18] proposed the principal components analysis (PCA) algorithm to reduce the dimension of the data, which effectively improved the frequency estimation accuracy. Based on the low rank characteristics of the signal matrix, ref [19] proposed the nuclear norm to restore the signal matrix, which improved the anti-noise performance of the algorithm. Although the above methods were able to reduce the influence of noise on the accuracy of frequency estimation, they were unable to distinguish signals with close frequencies.

In this paper, an enhanced root-MUSIC algorithm that combined the matrix reconstruction with the root-MUSIC algorithm were proposed. The frequency components of the signal were independent of each other, so the signal matrix was a low-rank matrix. The rank of the matrix was equal to the number of non-zero singular values of the matrix, from which the singular value vector of the signal matrix was sparse. Under the influence of noise, the singular value vector of the signal matrix was no longer sparse. In order to restore the sparsity of singular value vectors, a convex optimization problem was formulated to optimize the singular value vector. In order to obtain the singular value vector, we proposed two algorithms to solve this optimization problem according to whether the standard deviation of noise needed to be estimated or not. One algorithm was used to estimate the standard deviation of noise, and the other solved the optimization problem directly without the standard deviation of noise estimation. The signal matrix was reconstructed by utilizing the optimized singular value vector. Since the anti-diagonal elements of the Hankel matrix were equal, the Hankel structure of the signal matrix was restored by averaging the anti-diagonal elements. Then, the conventional root-MUSIC algorithm was utilized to estimate the frequencies of the signals based on the eigenvalue decomposition of the covariance matrix constructed from this reconstructed signal matrix. Finally, the simulation results showed that the proposed algorithm had a more accurate frequency estimation and a better frequency resolution for multi-frequency signals, than existing algorithms, in a noisy environment.

This paper is organized as follows. Section 2 introduces the signal model for the FMCW radar range and the frequency estimation method, based on the root-MUSIC algorithm. The enhanced root-MUSIC algorithm based on matrix reconstruction is proposed in Section 3. In Section 4, the performance of the algorithm is simulated and analyzed. The conclusion is provided in Section 5.

## 2. Signal Model Furthermore, Related Works

In this section, the signal model is firstly described. Then, the frequency estimation algorithm based on the root-MUSIC algorithm is described.

### 2.1. Signal Model

The transmitted FMCW signal can be expressed, as follows [20]: (1)$$s_{tx}\left(t\right)=A_{tx}\exp\left(j2\pi \left(f_{0}t+\frac{\xi }{2}t^{2}\right)+\varphi_{tx}\right),$$
where $A_{tx}$ is the amplitude of the transmitted signal, $\varphi_{tx}$ is the initial phase, $f_{0}$ is the initial frequency, ξ is the slope of the FMCW chirp, $\xi =B/T_{c}$, *B* is the bandwidth, and $T_{c}$ is modulation period.

The transmitted signal is reflected by the target to obtain the received signal, as follows [20]: (2)$$s_{rx}\left(t\right)=A_{rx}\exp\left(j2\pi \left(f_{0}\left(t-\tau \right)+\frac{\xi }{2}{\left(t-\tau \right)}^{2}\right)+\varphi_{rx}\right),$$
where $A_{rx}$ is the amplitude of the received signal, $\varphi_{rx}$ is the phase of the received signal, τ is the delay between the transmitted FMCW signal and the received signal, τ=2R/c, *R* is the distance between the target and radar, and *c* is the speed of electromagnetic waves.

After mixing the transmitted signal and the received signal and then conducting a low-pass filtering of the mixed signal, the obtained signal beat frequency can be expressed, as in [20]: (3)$$s\left(t\right)=\frac{A_{tx}A_{rx}}{2}\exp\left(j(2\pi \xi t\tau +2\pi f_{0}\tau -\pi \xi \tau^{2}-\varphi_{rx}+\varphi_{tx})\right).$$

Since τ is small, the term of $\tau^{2}$ can be ignored, so the signal beat frequency is expressed as: (4)$$s\left(t\right)=\frac{A_{tx}A_{rx}}{2}\exp\left(j(2\pi \xi t\tau +2\pi f_{0}\tau -\varphi_{rx}+\varphi_{tx})\right)=\frac{A_{tx}A_{rx}}{2}e^{j\psi (t)},$$
where ψ(t) is the phase of the signal beat frequency, $\psi \left(t\right)=2\pi \xi t\tau +2\pi f_{0}\tau -\varphi_{rx}+\varphi_{tx}$, $f_{c}$ is the signal beat frequency, and $f_{c}=\frac{1}{2\pi }\frac{d\psi (t)}{dt}=\xi \tau$.

Figure 1 shows the principle of frequency offset generated by distance. According to τ=2R/c, the distance between the target and the radar can be expressed as: (5)$$R=\frac{c\cdot f_{c}}{2\xi }.$$

### 2.2. Frequency Estimation Algorithm Based on the Root-MUSIC Algorithm

When FMCW radar is used for the range in the case of multiple targets, the signal beat frequency of the radar is composed of multiple frequency components. Assuming there are *D* targets, the discrete signal beat frequency can be expressed as: (6)$$s\left(n\right)=\sum_{i=1}^{D}A_{i}\exp\left(j\left(2\pi \frac{f_{i}n}{f_{s}}+\varphi_{i}\right)\right),n=0,1,\cdots ,P-1,$$
where $A_{i}$, $\varphi_{i}$ and $f_{i}$ are, respectively, the amplitude, phase and frequency of the signal beat corresponding to the ith target, $f_{s}$ is the sampling frequency, and *P* is the number of samples.

When considering Gaussian white noise, the signal beat frequency can be expressed as: (7)x(n)=s(n)+z(n),
where z(n) is Gaussian white noise with zero mean and variance $\sigma^{2}$.

In order to use the root-MUSIC algorithm for frequency estimation, we used x(n) to construct a M×L Hankel matrix X: (8)$$X=\left[\begin{array}{cccc} x(0) & x(1) & \cdots & x(L-1)\\ x(1) & x(2) & \cdots & x(L)\\ \vdots & \vdots & \ddots & \vdots \\ x(M) & x(M+1) & \cdots & x(P-1) \end{array}\right],$$
where L=P−M+1.

Then, *X* can be expressed as: (9)X=AS+N,
where $A=\left[a\left(w_{1}\right),\cdots ,a\left(w_{D}\right)\right]$, $a\left(w_{i}\right)={\left[\begin{array}{cccc} 1 & \exp(w_{i}j) & \cdots & \exp(w_{i}j\left(M-1\right)) \end{array}\right]}^{T}$ is the frequency vector, $w_{i}=2\pi f_{i}/f_{s}$, $S={\left[s_{1},\cdots ,s_{D}\right]}^{T}$, $s_{i}={\left[\begin{array}{cccc} A_{i}\exp\left(j\varphi_{i}\right) & A_{i}\exp\left(j\left(w_{i}+\varphi_{i}\right)\right) & \cdots & A_{i}\exp\left(j\left(w_{i}\left(L-1\right)+\varphi_{i}\right)\right) \end{array}\right]}^{T}$, and *N* is the noise matrix.

The covariance matrix $R_{X}$ is calculated from matrix *X* and can be expressed as: (10)$$R_{X}=E\left[XX^{H}\right]=AR_{S}A^{H}+\sigma^{2}I,$$
where $R_{S}=E\left[SS^{H}\right]$ is the covariance matrix of the matrix *S*, and $\sigma^{2}$ is the variance of noise.

The eigenvalue decomposition of the covariance matrix can be used to obtain the signal subspace $E_{S}$ and the noise subspace $E_{N}$.
(11)$$R_{X}=\left[\begin{array}{cc} E_{S} & E_{N} \end{array}\right]\left[\begin{array}{cccc} \lambda_{1} & 0 & \cdots & 0\\ 0 & \lambda_{2} & \ddots & \vdots \\ \vdots & \vdots & \ddots & 0\\ 0 & 0 & 0 & \lambda_{M} \end{array}\right]\left[\begin{array}{c} E_{S}^{H}\\ E_{N}^{H} \end{array}\right],$$
where $\lambda_{1},\cdots \lambda_{D}$ is the eigenvalues of the signal subspace $E_{S}$, and $\lambda_{D+1}=\lambda_{D+2}=\cdots =\lambda_{M}=\sigma^{2}$ is the eigenvalues of the noise subspace $E_{N}$.

Since the signal subspace and the noise subspace are orthogonal, we knew that
(12)$$A^{H}E_{N}=O.$$

Based on (12), we derived
(13)$$a{\left(w\right)}^{H}E_{N}=0,w=w_{1},w_{2}\cdots w_{D}.$$

Based on (13), we constructed the polynomial
(14)$$f\left(w\right)=p^{H}\left(w\right)E_{N}E_{N}^{H}p\left(w\right)=0,$$
where p(w)=[1,exp(−jw),⋯exp(−jw(M−1))].

After solving (14), the estimate ${\hat{w}}_{i}$ can be obtained. Then, the signal frequency can be estimated as: (15)$${\hat{f}}_{i}=\frac{f_{s}}{2\pi }{\hat{w}}_{i}.$$

Based on the relationship between frequency and distance in (5), the distance between target and radar was estimated as follows: (16)$${\hat{R}}_{i}=\frac{c\cdot {\hat{f}}_{i}}{2\xi }.$$

To summarize, the frequency estimation algorithm based on the root-MUSIC algorithm was summarized in Algorithm 1.
**Algorithm 1** Frequency Estimation Algorithm Based on the Root-MUSIC Algorithm**Require:** Original noisy signal x(n),n=0,1,⋯,P−1 Dimension of the Hankel matrix *M* and *L* (L=P−M+1) The number of the target *D***Ensure:** the distance between target and radar R^i 1. Construct a Hankel matrix *X* by (9) 2. Calculate the covariance matrix of the reconstructed matrix: RX=E[XXH] 3. Take eigenvalue decomposition for RX and obtain EN 4. Solve f(w)=pH(w)ENENHp(w)=0 5. Calculate f^i=fs2πwi,i=1,2,⋯,D 6. Calculate R^i=c·f^i2ξ,i=1,2,⋯,D

## 3. The Proposed Enhanced Root-MUSIC Algorithm Based on Matrix Reconstruction for Frequency Estimation

In this section, the optimization problem for the enhanced root-MUSIC algorithm based on matrix reconstruction is formulated. Then, we propose two algorithms to solve this optimization problem.

### 3.1. Optimization Problem Formulation for Enhanced Root-MUSIC Algorithm Based on Matrix Reconstruction

When the noise in the matrix *X* is relatively strong or the number of samples *P* is small, the frequency resolution of the root-music algorithm are significantly reduced, and it is difficult to distinguish the close frequencies. In order to improve the performance of the root-MUSIC algorithm, we decided to reconstruct the matrix *X* to mitigate the noise in *X*.

The matrix $S_{1}$ can be obtained by stacking s(n).
(17)$$S_{1}=\left[\begin{array}{cccc} s(0) & s(1) & \cdots & s(L-1)\\ s(1) & s(2) & \cdots & s(L)\\ \vdots & \vdots & \ddots & \vdots \\ s(M) & s(M+1) & \cdots & s(P-1) \end{array}\right]=\mathrm{AS}.$$

Obviously, $S_{1}$ is a low-rank Hankel matrix, and its rank is *D*. The singular value decomposition (SVD) of $S_{1}$ can be expressed as: (18)$$S_{1}=U_{S}\Sigma_{S}{V_{S}}^{H},$$
where $U_{S}$ and $V_{S}$ are unitary matrix, $\Sigma_{S}$ is a diagonal matrix, and $\Sigma_{S}=diag(\sigma_{1},\sigma_{2},$$\cdots \sigma_{D},0,\cdots ,0)$.

Based on matrix $\Sigma_{S}$, the singular value vector can be obtained as: (19)$$l=\left[\sigma_{1},\sigma_{2},\cdots ,\sigma_{D},0,\cdots ,0,0\right].$$

When considering the noise, the SVD in (8) can be expressed as: (20)$$X=U_{X}\Sigma_{X}{V_{X}}^{H},$$
where $U_{X}$ and $V_{X}$ are unitary matrix, $\Sigma_{X}$ is diagonal matrix, and $\sum_{X}=diag\left(\sqrt{{\sigma_{1}}^{2}+\sigma^{2}},\sqrt{{\sigma_{2}}^{2}+\sigma^{2}},\cdots ,\sqrt{{\sigma_{D}}^{2}+\sigma^{2}},\sigma_{D+1},\cdots ,\sigma_{M}\right)$, $\sigma_{D+1}=\sigma_{D+2}=\cdots =\sigma_{M}=\sigma$.

Based on $\sigma_{D+1}=\sigma_{D+2}=\cdots =\sigma_{M}=\sigma$, the singular value vector was constructed as follows: (21)$$\begin{array}{c} t=\left[\sigma_{1},\sigma_{2},\cdots ,\sigma_{D},\sigma_{D+1}-\sigma_{M},\cdots ,\sigma_{M-1}-\sigma_{M},0\right]\\ \;\;=\left[\sqrt{{\sigma_{1}}^{2}+\sigma^{2}},\sqrt{{\sigma_{2}}^{2}+\sigma^{2}},\cdots ,\sqrt{{\sigma_{D}}^{2}+\sigma^{2}},0,\cdots ,0\right]. \end{array}$$

At this time, there was an error of between the singular value vector t and the theoretical value I.
(22)$$\begin{array}{cc} {∥t-l∥}_{2} & =\sqrt{\sum_{i=1}^{D}{\left(\sqrt{{\sigma_{i}}^{2}+\sigma^{2}}-\sigma_{i}\right)}^{2}}\\ \; & =\sqrt{\sum_{i=1}^{D}(2{\sigma_{i}}^{2}+\sigma^{2}-2\sigma_{i}\sqrt{{\sigma_{i}}^{2}+\sigma^{2}})}\\ \; & \leq \sqrt{\sum_{i=1}^{D}\left(\sigma^{2}\right)}=\sqrt{D}\sigma . \end{array}$$

Due to the limited number of samples, there was an error between the obtained singular value and the theoretical value. At this time, $\sigma_{D+1}\geq \sigma_{D+2}\geq \cdots \geq \sigma_{M}$. Therefore, t can be expressed, as follows: (23)$$t=\left[\sigma_{1},\sigma_{2},\cdots ,\sigma_{D},\sigma_{D+1}-\sigma_{M},\cdots ,\sigma_{M-1}-\sigma_{M},0\right].$$

From (23), we found that the singular value vector was no longer a sparse vector. Based on the sparsity of the singular value vector, we formulated a convex optimization problem to reconstruct the singular value vector of the matrix, which restored the low rank properties of the matrix and reduced the interference of noise on the matrix. To achieve this aim, this convex optimization problem was formulated, as follows: (24)$$\begin{array}{c} \min{∥l∥}_{0}\\ s.t.{∥t-l∥}_{2}\leq \beta , \end{array}$$
where β is used to constrain the influence of noise on the singular values, and $\beta =\sqrt{D}\sigma$.

We obtained the optimal singular value vector l by solving the convex optimization problem (24), and the solution methods are proposed in the next subsection. Based on the estimated l, the matrix X was reconstructed, as follows: (25)$$X_{1}=U_{X}\Sigma_{X1}{V_{X}}^{H},$$
where $\Sigma_{X1}=diag\left(l\right)$.

However, the reconstructed matrix $X_{1}$ was no longer a Hankel matrix. In order to restore the Hankel structure of the matrix $X_{1}$, the elements on the anti-diagonal of the matrix were obtained by averaging the anti-diagonal elements of $X_{1}$.
(26)$$X_{2}\left(i,j\right)=\frac{\sum_{i+j=k,i=1}^{k-1}X_{1}\left(i,j\right)}{k-1},k=2,3,...,P+1,i+j=k.$$

Following the procedure from (8) to (16), the conventional root-MUSIC algorithm was then applied to the reconstructed matrix $X_{2}$ to estimate the distance between the target and the radar.

### 3.2. Enhanced Root-MUSIC Algorithm Based on Matrix Reconstruction

The convex optimization problem (24) was NP-hard [21]. Since $l_{1}$ norm was the optimal convex approximation of the $l_{0}$ norm [21,22], (24) was relaxed to solve the $l_{1}$ norm minimization problem, as follows: (27)$$\begin{array}{c} \min{∥l∥}_{1}\\ s.t.{∥t-l∥}_{2}\leq \beta . \end{array}$$

Theoretically, $\beta =\sqrt{D}\sigma$. To solve (27), we proposed two solution methods according to whether the standard deviation of noise needs to be estimated. The first solution method required the standard deviation of noise σ. When σ was known, we directly solved (27) to estimate the optimal singular value vector l. However, the standard deviation of noise σ was unknown in most cases. Therefore, we firstly estimated σ as: (28)$$\hat{\sigma }=\sqrt{\frac{\sum_{i=D+1}^{M}{\sigma_{i}}^{2}}{M-D}}.$$

Based on (21), the energy of the noise was distributed across all singular value vectors. Because the signal length was limited, there was an error between the obtained singular value and the theoretical value. At this time, $\sigma_{1}\geq \sigma_{2}\geq \cdots \geq \sigma_{D}\geq \sigma_{D+1}\geq \cdots \geq \sigma_{M}$. When the standard deviation of the noise was estimated by (29), the noise energy distributed on the first D singular values could not be estimated, resulting in loss and making the estimated value less than the actual value.

In order to compensate for the estimation error of the standard deviation of the noise σ^, we considered that β=kDσ^,k>1, where *k* is a compensation coefficient. We used CVX toolbox [23] in MATLAB to solve (27) and obtain the optimal singular value vector l. The proposed enhanced root-MUSIC algorithm obtained by (27) added the signal matrix reconstruction process between the first and second steps of Algorithm 1. The specific steps were summarized in Algorithm 2.
**Algorithm 2** Enhanced Root-MUSIC algorithm via solution of (27)**Require:** Original noisy signal x(n),n=0,1,⋯,P−1 Dimension of the Hankel matrix *M* and *L* (L=P−M+1) The number of the target *D***Ensure:** the distance between target and radar R^i 1. Construct a Hankel matrix *X* by (8) 2. Take singular value decomposition for *X*; thus, UX, VX and ΣX can be obtained by using (20) 3. Construct the singular value matrix t, t=σ1,σ2,⋯,σD,σD+1−σM,⋯,σM−1−σM,0 4. Estimate the standard deviation of noise σ^ 5. Use CVX toolbox to solve (27) to obtain the optimal singular value vector l 6. Let ΣX1=diag(l), and reconstruct the signal matrix X1=UXΣX1VXH 7. Reconstruct the Hankel matrix by (26) 8. Calculate the covariance matrix of the reconstructed matrix: RX=E[X2X2H] 9. Perform steps 3–6 of Algorithm 1

To solve (24) using the above method, the standard deviation σ of the noise was required. When the standard deviation σ of the noise was known, the above methods effectively estimated the singular value vector l. However, when the standard deviation σ of the noise was not known and needed to be estimated, the estimation performance of Algorithm 2 degraded due to the estimation error of the standard deviation σ of the noise.

In order to reduce the impact of σ on the estimation performance of Algorithm 2, we utilized the Lagrange multiplier method to solve (27), and the optimization problem in (27) was reformulated as: (29)$$\min\lambda {∥l∥}_{1}+{∥t-l∥}_{2},$$
where λ is a regularization parameter, which is used to control the sparsity of the singular value vector.

In this case, the standard deviation σ of noise estimate was not needed. We used the CVX toolbox [23] in MATLAB to solve (29) in order to obtain the optimal singular value vector l. The proposed enhanced root-MUSIC algorithm obtained by(29) added the signal matrix reconstruction process between the first and second steps of Algorithm 1. The specific steps were summarized in Algorithm 3.

To summarize, the proposed enhanced root-MUSIC algorithm based on matrix reconstruction for estimating the signal beat frequency in FMCW radar had two solution methods, both of which reduced the effect of the noise on the frequency estimation accuracy. Therefore, the proposed enhanced root-MUSIC algorithms had better estimation performance than the standard root-MUSIC algorithm.
**Algorithm 3** Enhanced Root-MUSIC algorithm via solving (27)**Require:** Original noisy signal x(n),n=0,1,⋯,P−1 Dimension of the Hankel matrix *M* and *L* (L=P−M+1) The number of the target *D***Ensure:** the distance between target and radar R^i 1. Construct a Hankel matrix *X* by (8) 2. Take singular value decomposition for *X*; thus, UX, VX and ΣX can be obtained by using (21) 3. Construct the singular value matrix t, t=σ1,σ2,⋯,σD,σD+1−σM,⋯,σM−1−σM,0 4. Use CVX toolbox to solve (29) in order to obtain the optimal singular value vector l 5. Let ΣX1=diag(l), and then reconstruct the signal matrix X1=UXΣX1VXH 6. Reconstruct the Hankel matrix by (26) 7. Calculate the covariance matrix of the reconstructed matrix: RX=E[X2X2H] 8. Perform steps 3–6 of Algorithm 1

## 4. Simulations and Analysis

In this section, the simulation results are presented to verify the performance of the proposed enhanced root-MUSIC algorithms (Algorithms 2 and 3). Firstly, we compared the above two algorithms with the root-MUSIC algorithm (Algorithm 1). Then, we compared the proposed algorithms with the existing methods, including the FFT algorithm, the CZT algorithm [24], the MUSIC algorithm, the ESPRIT algorithm, the PCA-MUSIC algorithm [18], and the WNNM ESPRIT algorithm [19]. The performance of all algorithms was evaluated by 2000 independent Monte Carlo iterations for each signal-to-noise ratio (SNR) in MATLAB. In all test cases, the simulation parameters were set as shown in Table 1.

The root-mean-square error (RMSE) was used to verify the performance of the algorithm. It was expressed, as follows: (30)$$RMSE=\sqrt{\frac{\sum_{I=1}^{K}{\left({\hat{R}}_{i}-R_{i}\right)}^{2}}{K}}.$$

### 4.1. Performance Comparison of Two Enhanced Root-MUSIC Algorithms

We compared the performance of the two enhanced root-MUSIC algorithms (Algorithms 2 and 3) with the conventional root-MUSIC algorithm (Algorithm 1) for the single-target scenario and the multi-target scenario, respectively. The essence of the FMCW radar range is the estimation of the signal beat frequency. When setting the simulation parameters, we first determined the frequency of the signal and then calculate the theoretical range value, which led to the fact that the distance between the target and the radar was not an integer. We set the signal beat frequency to 5500 Hz, and the corresponding distance between the radar and the target was 9.1011 m. Considering that in a multi-target scenario, in addition to the targets with longer distances, there were likely to be targets with similar distances, three frequency components of the signals were set. Based on the sampling frequency and the signal lengths determined in Table 1, we calculated the frequency resolution of the FFT algorithm at 90.6089 Hz. In order to enable the FFT algorithm to distinguish similar frequency components, we set the frequency differences of similar frequency components at 100 Hz. We set the signal beat frequency of the target nearest to the radar at 5100 Hz, and the other target’s signal beat frequency was 5200 Hz.The distances corresponding to the two signal beat frequencies were 8.4392 m and 8.6047 m, respectively. By using these signal frequency settings, we believed that the performance of the various algorithms could be fully verified.

In a single-target scenario, we assumed that the distance between the target and the radar was 9.1011 m while the signal beat frequency was 5500 Hz. Figure 2 shows the RMSE values of the distance estimations versus the SNR values of all three algorithms when R=9.1011 m.

As shown in Figure 2, the RMSE values of the two enhanced root-MUSIC algorithms (Algorithms 2 and 3) were smaller than that of the root-MUSIC algorithm under different SNR values, and the two enhanced root-MUSIC algorithms (Algorithms 2 and 3) had similar performances in a single-target scenario.

In a multi-target scenario, we assumed that the distances between the target and the radar were 8.4392 m, 8.6047 m and 9.1011 m, while the signal beat frequency was 5100 Hz, 5200 Hz and 5500 Hz, respectively. Figure 3 shows the RMSE values of distance estimations versus the SNR values of all three algorithms when R=8.4392 m. Figure 4 and Figure 5 show the corresponding results when R=8.6047 m and R=9.1011 m, respectively.

As shown in Figure 3, Figure 4 and Figure 5, the RMSE values of the two enhanced root-MUSIC algorithms (Algorithms 2 and 3) were much smaller than that of the root-MUSIC algorithm (Algorithm 1) under different SNR values, which indicated the proposed algorithms significantly improved the frequency resolution of multi-frequency signals in a noisy environment. There were signal components with similar frequencies that were affected by the noise, and the signal components with similar frequencies were, at times, detected as a single component. The noise, at times, was wrongly detected as a signal component, which resulted in the performances shown in Figure 3 and Figure 5 being worse than those in Figure 4. When the SNR value was larger than 15, the root-MUSIC algorithm (Algorithm 1) no longer detected the noise as a signal component. Therefore, the frequencies corresponding to the three signal components were then correctly estimated, so the performance was greatly improved. When the SNR value was higher than 12 dB, the proposed Algorithm 2 was basically identical with the proposed Algorithm 3. In general, the proposed Algorithm 3 had better anti-noise performance than the proposed Algorithm 2.

In summary, the range accuracy of the two enhanced root-MUSIC algorithms (Algorithms 2 and 3) was higher than that of the conventional root-MUSIC algorithm (Algorithm 1). In a single-target scenario, the proposed two algorithms (Algorithms 2 and 3) had similar performances. In a multi-target scenario, the proposed two enhanced root-MUSIC algorithms (Algorithms 2 and 3) were able to distinguish between the targets with close ranges while the conventional root-MUSIC (Algorithm 1) had a large error in its range values. Compared with Algorithm 2, Algorithm 3 was less affected by the noise. Overall, proposed Algorithm 3 had the highest range accuracy.

### 4.2. Performance Comparison of Existing Algorithms

#### 4.2.1. Estimation Performance for a Single-Target Scenario

In some applications, radar is applied to single-target ranges. In this case, the signal beat frequency is a single-frequency signal. We assumed that the distance between the target and the radar was 9.1011 m while the signal beat frequency was 5500 Hz. Figure 6 shows the RMSE values of the distance estimations versus the SNR values of various algorithms when R=9.1011 m.

As shown in Figure 6, the RMSE values of the distance estimations of proposed Algorithms 2 and 3 were smaller than those estimated by other algorithms under different SNR values. Due to the fence effect, the RMSE values of the distance estimations of the FFT algorithm and the chirp-Z transform (CZT) algorithm were basically stable. Similar to the frequency-domain method, the performances of the MUSIC algorithm and its improved algorithm were related to search times. With increased search times, the performance of the MUSIC algorithm and its improved algorithm were improved, but it increased the computational complexity. The performances of the proposed Algorithms 2 and 3 were better than those of the ESPRIT algorithm and its improved algorithm in a single-target scenario. Overall, the proposed Algorithms 2 and 3 had the best performances in a single-target scenario.

#### 4.2.2. Estimation Performance for a Multi-Target Scenario

In this simulation, we assumed that there were three targets that needed to be measured at the same time. The distances between the targets and the radar were 8.4392 m, 8.6047 m and 9.1011 m while the corresponding frequencies were 5100 Hz, 5200 Hz and 5500 Hz, respectively. Figure 7 shows the RMSE values of the distance estimations versus the SNR values of the various algorithms when R=8.4392 m. Figure 8 and Figure 9 show the corresponding results when R=8.6047 m and R=9.1011 m, respectively.

As shown in Figure 7 and Figure 8, the RMSE values of the MUSIC algorithm, the ESPRIT algorithm and their improved algorithms were larger than other algorithms under low SNR values. Since these were affected by the noise, there were large errors in the estimations of the signal subspace and the noise subspace, which meant it could not distinguish close frequencies. Similar to a single-target scenario, the RMSE values of the distance estimations of the FFT algorithm and the CZT algorithm were stable. As shown in Figure 7 and Figure 8, the performance of proposed Algorithm 2 was better than the CZT algorithm when the SNR value was larger than 10 dB. In Figure 9, the RMSE value of proposed Algorithm 2 was smaller than that of the CZT algorithm. Overall, the proposed algorithms (Algorithms 2 and 3) and the CZT algorithm had better performances. However, the CZT algorithm still had the problem of the fence effect, so its performance was not improved when as the SNR values increased. The proposed algorithms (Algorithms 2 and 3) had the problem of high computational complexity, which indicated they could not be applied in real-time applications. For a scenario with a high range accuracy, proposed Algorithm 3 was the best choice. The RMSE values of the distance estimations by proposed Algorithm 3 were the smallest under different SNR values, which indicated proposed Algorithm 3 could estimate close frequencies effectively.

## 5. Conclusions

In this paper, an enhanced root-MUSIC algorithm based on matrix reconstruction was proposed. Based on the sparsity of the singular value vector, a convex optimization problem was formulated to optimize the singular value vector. We proposed two different algorithms to solve the convex optimization problem. The signal matrix was reconstructed using the estimated singular value vector, and the Hankel structure of the signal matrix was restored by utilizing the properties of the Hankel matrix. Then, the covariance matrix was calculated by the reconstructed matrix, and the conventional root-MUSIC algorithm was utilized to estimate the signal beat frequency based on the eigenvalue decomposition of the covariance matrix. The simulation results showed that the performances of the two proposed algorithms were much better than the traditional root-MUSIC algorithm. Compared with other algorithms, the proposed algorithms still had more accurate frequency estimations and better frequency resolutions in the presence of multi-frequency signals in a noisy environment, which improved the multi-target range accuracy and resolution capabilities of FMCW radar. However, the algorithms proposed in this paper had high computational demands, which is the disadvantage of subspace-based algorithms. In the future, we will consider reducing the amount of computation by down-sampling the signal matrix and optimizing the calculations of the singular-value and eigenvalue decompositions, which has practical significance for expanding the application of subspace-based algorithms. 

## Figures and Tables

**Figure 1 sensors-23-01829-f001:**
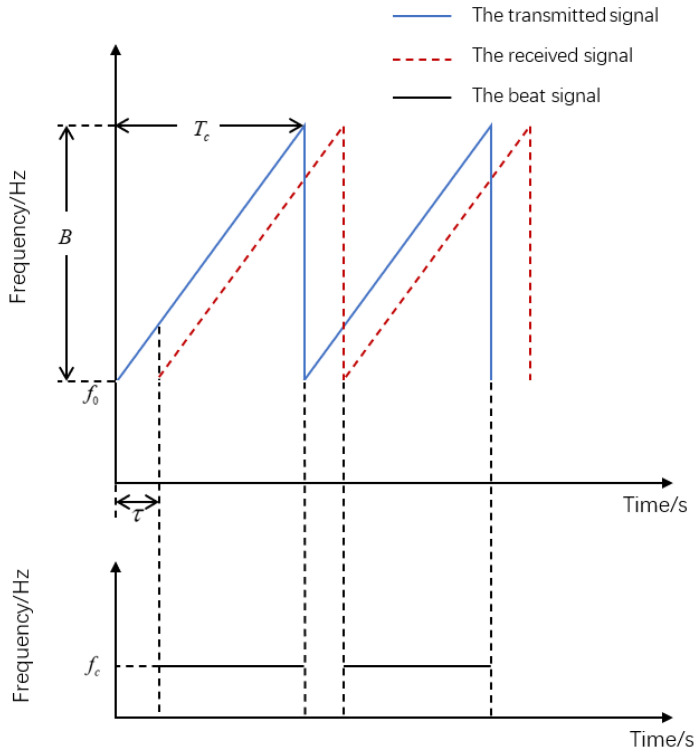
Schematic diagram of frequency offset generated by distance.

**Figure 2 sensors-23-01829-f002:**
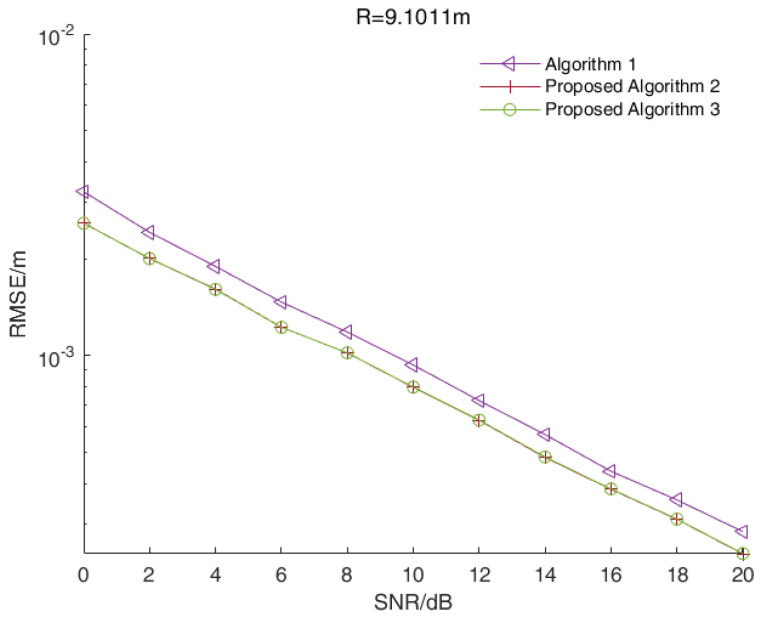
The RMSE values of the distance estimations versus the SNR values of all three algorithms in a single-target scenario.

**Figure 3 sensors-23-01829-f003:**
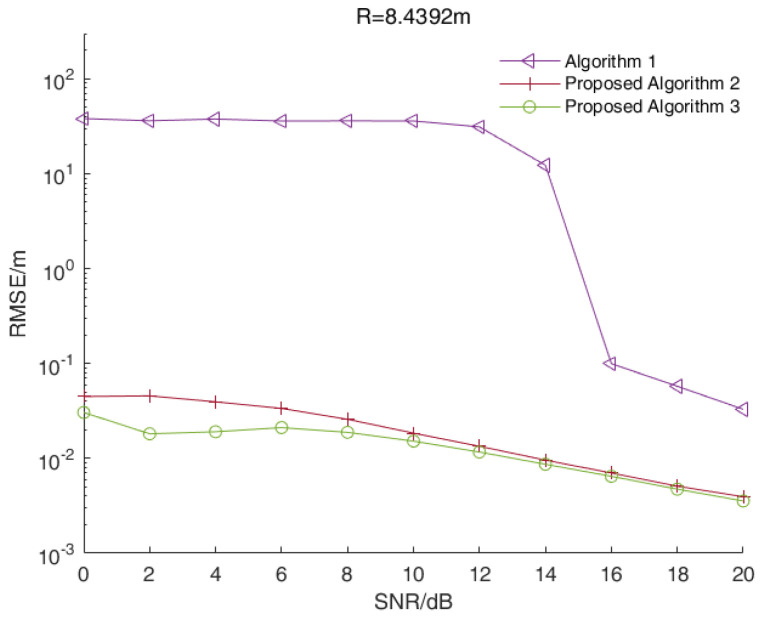
The RMSE values of the distance estimations versus the SNR values of all three algorithms when R=8.4392 m.

**Figure 4 sensors-23-01829-f004:**
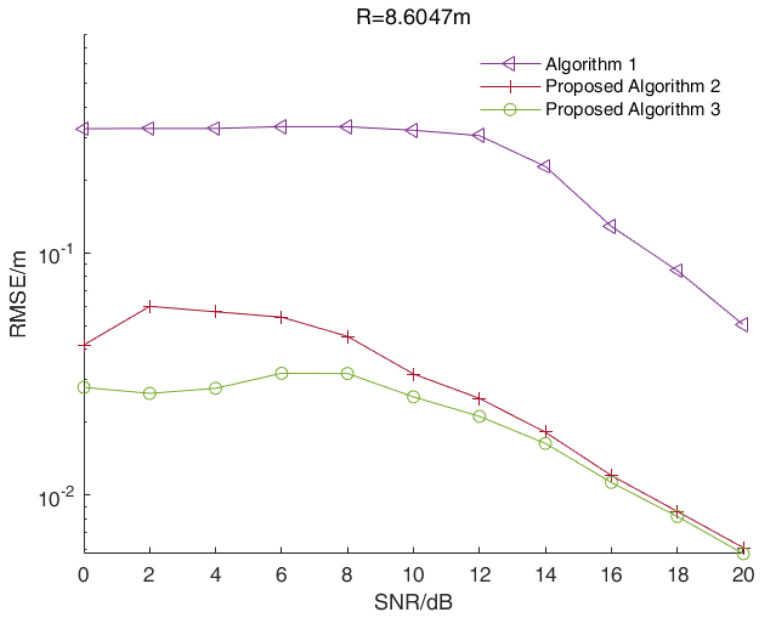
The RMSE values of the distance estimations versus the SNR values of all three algorithms when R=8.6047 m.

**Figure 5 sensors-23-01829-f005:**
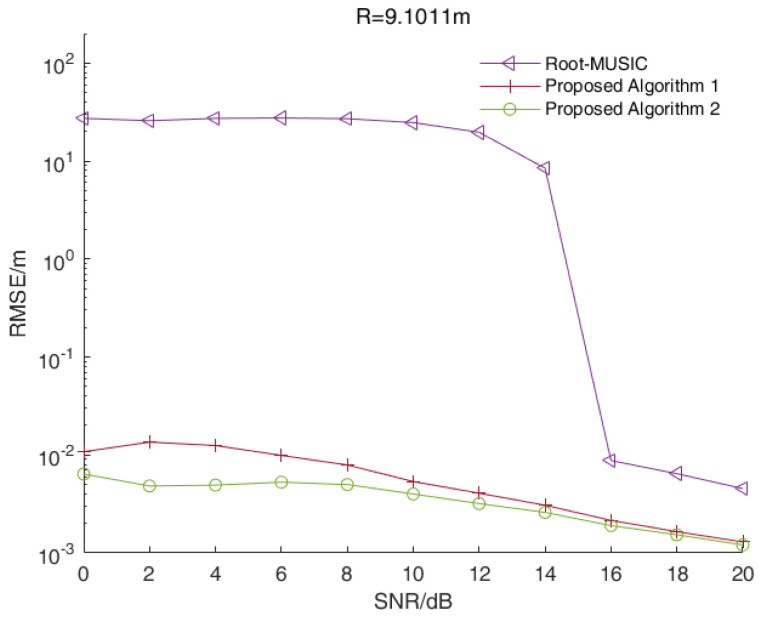
The RMSE values of the distance estimations versus the SNR values of all three algorithms when R=9.1011 m.

**Figure 6 sensors-23-01829-f006:**
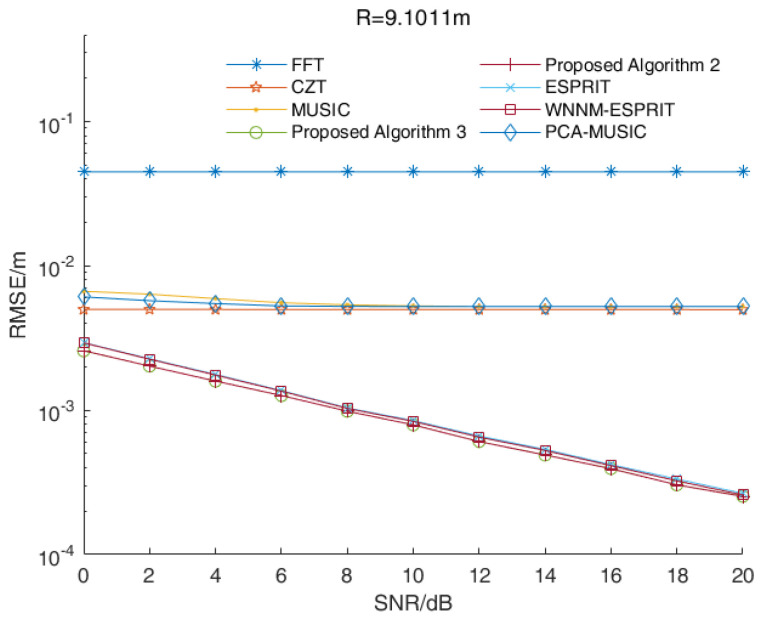
The RMSE values of the distance estimations versus the SNR values of various algorithms in a single-target scenario.

**Figure 7 sensors-23-01829-f007:**
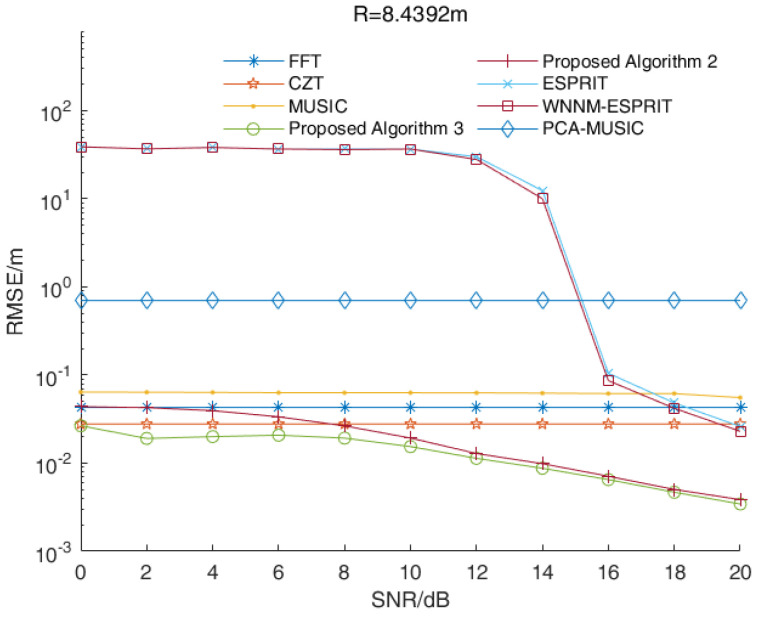
The RMSE values of the distance estimations versus the SNR values of various algorithms when R=8.4392 m.

**Figure 8 sensors-23-01829-f008:**
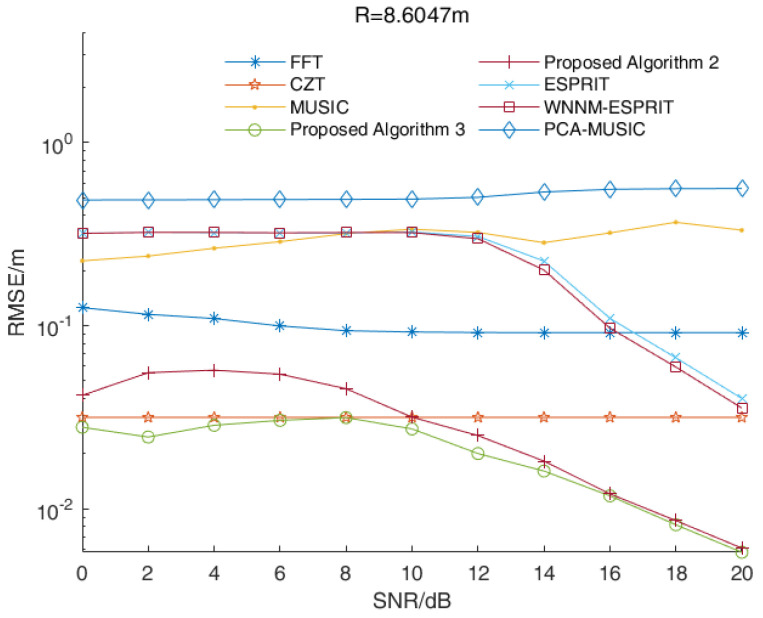
The RMSE values of the distance estimations versus the SNR values of various algorithms when R=8.6047 m.

**Figure 9 sensors-23-01829-f009:**
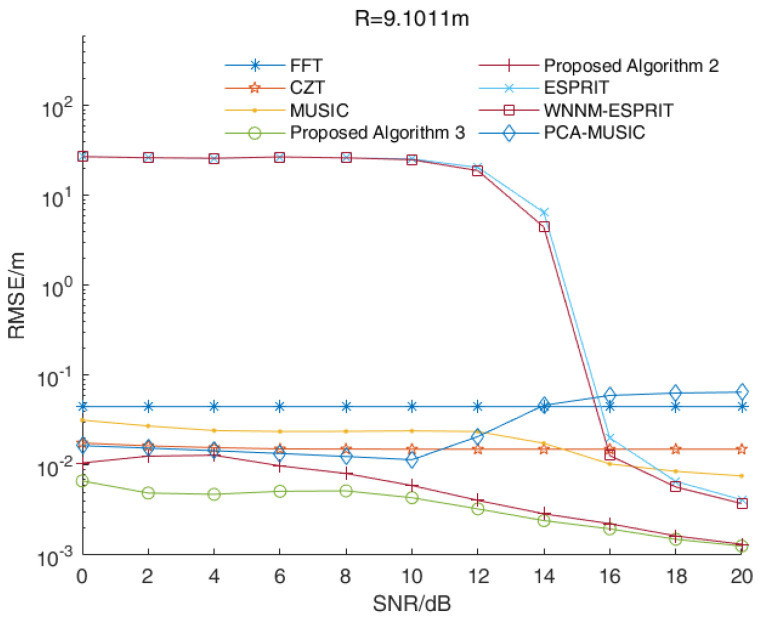
The RMSE values of the distance estimations versus the SNR values of various algorithms when R=9.1011 m.

**Table 1 sensors-23-01829-t001:** Simulations parameters.

Parameters	Value
Bandwidth *B*/MHz	999.47055
Speed of electromagnetic waves *c*/(km/s)	299709
Modulation period $T_{c}$/s	0.011
Sampling frequency $f_{s}$/kHz	92.7835
Initial frequency $f_{0}$/kHz	100
Signal length *P*	1024
SNR/dB	0–20
*M*	100
*L*	925
Multiple refinements of CZT algorithm	60
Search times of MUSIC algorithm	5000
Regularization parameter λ	0.6
Compensation coefficient *k*	3.3
Simulation times *K*	2000

## Data Availability

Not applicable.

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
