# Peer review of "Enhanced Root-MUSIC Algorithm Based on Matrix Reconstruction for Frequency Estimation"

_sensors, 2023, doi:10.3390/s23041829_

Round 1

Reviewer 1 Report

The authors propose two versions of enhanced a root-MUSIC algorithm for frequency-modulated continuous wave radar ranging applications. The paper is relatively well-written and the proposal is validated by means of numerical simulations.

For publication, I suggest addressing the following:

- Section 3.1 and 3.2 could be better organized to improve readability. First, it is important to highlight the contributions from the paper, telling them apart from the literature results. If I correctly understand, Sec. 3.1 describes the existing Root-MUSIC Algorithm and Sec. 3.2 describes the two proposed algorithms. As there are several steps mentioned in Sec. 3.1, I suggest presenting them as a Table and then, in Sec. 3.2, present only the necessary additional steps to solve the optimization problem.

- In Fig. 2, the curves for Algorithms 1 and 2 are overlapped and difficult to tell apart. I suggest changing the markers to ease visualization.

- In Figs. 6-9, is it not possible to evaluate Proposed Algorithm 1 as well?

- In Fig. 9, CZT performance drastically changes in comparison with the one from Figs. 7 and 8. Are there any ideas to explain why this happens?

- There are some minor issues when presenting some equations, such as (15) and (16), which should be followed by a period and not a comma. Besides that, the paper would  benefit from a careful proofread and language review to address typos and minor issues such as "MUSCI" in page 11.

Author Response

Point 1: Section 3.1 and 3.2 could be better organized to improve readability. First, it is important to highlight the contributions from the paper, telling them apart from the literature results. If I correctly understand, Sec. 3.1 describes the existing Root-MUSIC Algorithm and Sec. 3.2 describes the two proposed algorithms. As there are several steps mentioned in Sec. 3.1, I suggest presenting them as a Table and then, in Sec. 3.2, present only the necessary additional steps to solve the optimization problem.

Response 1: According to this comment, we have listed the frequency estimation process of the root-MUSIC algorithm (Algorithm 1) as a table in Sec. 3.1, and present only the necessary additional steps of the proposed enhanced root-MUSIC algorithm (Algorithm 2, algorithm 3) in section 3.2.

Point 2: In Fig. 2, the curves for Algorithms 1 and 2 are overlapped and difficult to tell apart. I suggest changing the markers to ease visualization.

Response 2: In the revised manuscript, the Algorithm 1 and 2 are changed to Algorithm 2 and Algorithm 3. The reason why the curves of Algorithm 2 and Algorithm 3 overlap is that the performances of the Algorithm 2 and Algorithm 3 are basically the same in the single target ranging scenario. According to the comment, we have changed the markers to ease visualization.

Point 3: In Figs. 6-9, is it not possible to evaluate Proposed Algorithm 1 as well?

Response 3: The proposed Algorithm 1 can be evaluated in Figs. 6-9. In the revised manuscript, we have added the performance of the proposed Algorithm 1 in Figs. 6-9.

Point 4: In Fig. 9, CZT performance drastically changes in comparison with the one from Figs. 7 and 8. Are there any ideas to explain why this happens?

Response 4: While using CZT algorithm, we need to determine the interval of the frequency refinement.

The frequency refinement interval we set before is about 360Hz. However, the frequency difference of the signal is 100Hz, and thus there may be two frequencies in the frequency refinement interval. In the process of the frequency estimation, the frequency estimation value of 5200Hz signal may be misjudged as the frequency estimation value of 5100Hz signal, resulting in the degradation of algorithm performance. In order to solve the above problems, we reset the frequency refinement interval. The frequency refinement interval is set between the maximum frequency estimate and the minimum frequency estimate, which can avoid misjudgment of the frequency estimate and improve the performance of the CZT algorithm. Based on this, we re-simulate the algorithms and modify Figure 7-9 in the revised manuscript.

Point 5: There are some minor issues when presenting some equations, such as (15) and (16), which should be followed by a period and not a comma. Besides that, the paper would benefit from a careful proofread and language review to address typos and minor issues such as "MUSCI" in page 11.

Response 5: We have carefully reviewed the contents of the manuscript, and revised and modified the existing problems and English usage in the manuscript.

Author Response

Point 1: The verb has been used in plural form which is wrong. There are also, many other mistakes which have to be revised very well in this paper.

Response 1: We have reviewed the contents of the paper, and revised the existing problems.

Point 2: In terms of technical points, there are many ambiguities. I believe that the authors have failed to explain them very well. Therefore, they have to answer these questions very well before they get their paper published.

First of all, what are the advantages of Frequency-Modulated Continuous Wave (FMCW) radar over

a simple continuous wave radar (CW-Radar)?

As shown in formula (5), the distance between the target and the radar can be expressed as:

On what circumstances, can the radar range be determined by a simple frequency comparison?

Response 2:

According to the working mode of radar, radar usually has two basic types: continuous wave (CW) radar and pulse radar. The continuous wave (CW) radar is a representative low complexity radar system, and uses only the difference between the carrier frequencies at the transmitter and receiver to estimate the velocity of the target. However, the CW radar cannot measure the distance to the target. Frequency-Modulated Continuous Wave (FMCW) radar is a continuous wave radar that uses a specific signal to modulate the frequency of the transmitted signal. The signal processing of the FMCW radar is performed in a low frequency band after mixing, and thus the FMCW radar systems are capable of estimating the range, Doppler, and angle of the targets.

For the FMCW system, the Range/Doppler migration and velocity ambiguity are two well-known problems encountered in high-speed moving target detection. Therefore, the range of the targets cannot be determined by a simple beat frequency estimate in the scenario of high-speed moving target detection. When there is no range migration, i.e., the target movement within a coherent processing interval (CPI) is less than a range resolution bin, the range of the targets can be determined by a simple beat frequency estimate.

Point 3: Furthermore, they have introduced an enhanced root-MUSIC algorithm based on matrix reconstruction but they have not discussed and reviewed other classical and modern methods of frequency estimation. Furthermore, it is necessary that a comparison be propounded which reveals some merits and demerits of the existing methods with the proposed method.

Response 3: In the first part, we add the description and review of other classical and modern frequency estimation methods. As introduced in section 1, there are basically two ways to estimate the frequencies of multi-frequency signal: time-domain methods and frequency-domain methods. We choose FFT algorithm and CZT algorithm as representatives of frequency-domain methods for comparison. The representative algorithms of time-domain methods include MUSIC algorithm, root-MUSIC algorithm, ESPRIT algorithm.

Considering that we reconstruct the signal matrix to improve the anti-noise performance of the algorithms, we select relevant improved algorithms for comparison, including PCA-MUSIC algorithm [19] and WNNM-ESPRIT algorithm [20]. In the section 4 and section 5, we describe the advantages and disadvantages of various algorithms in more detail.

Point 4: According to the lines 45 to 47, the authors claim that by analyzing the signal matrix, the singular value vector of the signal matrix is a sparse vector. But how? It needs more explanation.

Response 4: The frequency components of the signal are independent of each other, so the signal matrix is a low-rank matrix. The rank of the matrix is equal to the number of non-zero singular values of the matrix, from which we can know that the singular value vector of the signal matrix is sparse. Under the influence of noise, the singular value vector of signal matrix is no longer a sparse vector. In order to restore the sparsity of singular value vectors, a convex optimization problem is formulated to optimize the singular value vector.

Point 5: What is more, following the formula (28), the authors refer to the leakage of noise energy which causes

What kinds of signal leaks are available at this paper. The authors need to explain them very well.

Response 5: From the formula (20), we can see that the energy of noise is distributed in all singular value vectors. Because the signal length is limited, there is an error between the obtained singular value and the theoretical value. At this time,  . When the standard deviation of noise is estimated by formula (28), the noise energy distributed on the first D singular values cannot be estimated, resulting in loss and making the estimated value less than the actual value.

Point 6: Also, how did the authors assume some special numbers for the distance between the target and the radar, the frequency of beat signal for single target and multi-target scenarios?

Response 6: The essence of FMCW radar ranging is the frequency estimation of beat signal. When setting the simulation parameters, we first determine the frequency of the signal, and then calculate the theoretical range value, which leads to the fact that the distance between the target and the radar is not an integer. We set the frequency of the beat signal from the target farthest from the radar to 5500Hz. Considering that in the multi-target scenario, in addition to the targets with long distance, there will also be targets with similar distance, so three signal components are set. Based on the signal sampling frequency and signal length set by simulation, we can calculate the frequency resolution of FFT algorithm is 90.6089Hz. In order to enable FFT algorithm to distinguish similar frequency components, we set the frequency difference of similar frequency components to 100Hz. We set the frequency of the beat signal of the target nearest to the radar to 5100Hz, and the frequency of the other beat signal is 5200Hz. Through such signal frequency setting, we believe that the performance of the algorithm can be fully verified.

Point 7: Based on the figures 3, 4 and 5, as R increases, root-MUSIC graph, reveals totally different trends especially in figure 5, when SNR/dB is bigger than 15, there is a nose dive in the graph from almost 10^1 to 10 ^-2. To the best of my knowledge, it seems so weird.

Response 7: The Root-Music algorithm performs frequency estimation by solving polynomials. There are signal components with similar frequencies in the beat signal, affected by the noise, the signal components with similar frequencies may sometimes be estimated as a single component, and noise is sometimes may be wrongly detected as a signal component, which results in the performances in Figure 3 and Figure 5 being worse than that in Figure 4. When SNR/dB is bigger than 15, the root-MUSIC algorithm will no longer detect the noise as the signal components.  Therefore, the frequency estimation is corresponding to the estimation of three signal components, so the performance has been greatly improved.

Point 8: Also in Figure 8, the performance of MUSIC for the values of SNR/dB above 14 is erratic. Why? Following it, in Figure 9, the value of RMSE /m for the numbers of SNR /dB higher than 14 for R=9.1011, is totally different from that of Figure 8. How is it possible?

 Response 8: In the simulation, considering the amount of computation, the size of the constructed signal matrix is 100×925. However, this is not the best case for the performance of the subspace-based algorithms, when the signal matrix is 512×513, the performance of the subspace-based algorithms is the best, but this means that it needs more computation. The MUSIC algorithm is limited by the size of the signal matrix and thus does not converge. The signal matrix constructed in this paper is sufficient to verify the performance of our proposed algorithm, so we did not handle the unstable situation of the conventional MUSIC algorithm.

When the signal-to-noise ratio is 20dB, we draw the spatial spectrum of the MUSIC algorithm as shown in Figure. 1 below. Theoretically, the spatial spectrum should have three peaks, but it can be seen that in some cases, the spatial spectrum only has two peaks. We believe that the two peaks correspond to the frequency estimates of the 5100Hz and 5500Hz signals, while the corresponding to 5200Hz cannot be estimated. This situation leads to a large error in the frequency estimation of 5200Hz.

Figure 1 is in the word.

Reviewer 3 Report

In the review of the manuscript "Enhanced Root-MUSIC Algorithm Based on Matrix Reconstruction for Frequency Estimation", I found the work is good. However, how to validate the current study of proposed enhanced root-MUSIC algorithm based on matrix reconstruction?

Author Response

In section 4, we validate the performance of proposed enhanced Root-MUSIC algorithm based on matrix reconstruction by simulation. Firstly, we compare the proposed algorithms with the root-MUSIC algorithm, from which we can know that the proposed algorithms have better performance than that of Root-MUSIC algorithm. Then we compare the proposed algorithms with the existing algorithms. As introduced in section 1, there are basically two ways to estimate the frequencies of multi-frequency signal: time-domain methods and frequency-domain methods. We choose FFT algorithm and CZT algorithm as representatives of frequency-domain methods for simulation. The representative algorithms of time-domain methods include MUSIC algorithm, ESPRIT algorithm. Considering that the signal matrix is reconstructed to improve the anti-noise performance of the algorithms, we select two relevant existing algorithms for comparison, including PCA-MUSIC algorithm [19] and WNNM-ESPRIT algorithm [20]. In the simulation, we also consider the range scenarios of single target and multiple targets, and analyze them respectively, which comprehensively verified that the performances of the proposed algorithms outperform the existing algorithms. Overall, the proposed algorithms have better performances than other algorithms.

Round 2

Reviewer 2 Report

The changes  have  been  applied  to the  text